Immune infiltration and prognosis in gastric cancer: role of NAD+ metabolism-related markers

Xing Yu 1 2 3 4 5
Zhang Zili 1 2 3 4 5
Gao Wenqing 2 3 4 5
Song Weiliang 2 3 4 5
Li Tong 1 2 3 4 5 litong3zx@sina.com
1 The Third Central Clinical College of Tianjin Medical University , Tianjin , China
2 The Third Central Hospital of Tianjin , Tianjin , China
3 Tianjin Institute of Hepatobiliary Disease , Tianjin , China
4 Artificial Cell Engineering Technology Research Center , Tianjin , China
5 Tianjin Key Laboratory of Extracorporeal Life Support for Critical Diseases , Tianjin , China
Pfeffer Ulrich
Electronic publication date: 2024 Jul 31
Publication date: 2024
Volume: 12
Electronic Location ID: e17833
Received 2024 Apr 5; Accepted 2024 Jul 9
Copyright: © 2024 Xing et al.
Copyright year: 2024
Copyright holder: Xing et al.
License: This is an open access article distributed under the terms of the Creative Commons Attribution License, which permits unrestricted use, distribution, reproduction and adaptation in any medium and for any purpose provided that it is properly attributed. For attribution, the original author(s), title, publication source (PeerJ) and either DOI or URL of the article must be cited.
License URL: https://creativecommons.org/licenses/by/4.0/

Keywords: Gastric cancer, Nicotinamide adenine dinucleotide (NAD+) metabolism, Prognosis, Risk signature

Funding: Tianjin Key Medical Discipline (Specialty) Construction Project TJYXZDXK-035A Tianjin Science and Technology Project 21JCYBJC01590 This work was sponsored by the Tianjin Key Medical Discipline (Specialty) Construction Project (TJYXZDXK-035A) and the Tianjin Science and Technology Project (21JCYBJC01590). The funders had no role in study design, data collection and analysis, decision to publish, or preparation of the manuscript.

==============================
Background

This study endeavored to develop a nicotinamide adenine dinucleotide (NAD+) metabolism-related biomarkers in gastric cancer (GC), which could provide a theoretical foundation for prognosis and therapy of GC patients.

Methods

In this study, differentially expressed genes (DEGs1) between GC and paraneoplastic tissues were overlapped with NAD+ metabolism-related genes (NMRGs) to identify differentially expressed NMRGs (DE-NMRGs). Then, GC patients were divided into high and low score groups by gene set variation analysis (GSVA) algorithm for differential expression analysis to obtain DEGs2, which was overlapped with DEGs1 for identification of intersection genes. These genes were further analyzed using univariate Cox and least absolute shrinkage and selection operator (LASSO) regression analyses to obtain prognostic genes for constructing a risk model. Enrichment and immune infiltration analyses further investigated investigate the different risk groups, and qRT-PCR validated the prognostic genes.

Results

Initially, we identified DE-NMRGs involved in NAD biosynthesis, with seven (DNAJB13, CST2, THPO, CIDEA, ONECUT1, UPK1B and SNCG) showing prognostic significance in GC. Subsequent, a prognostic model was constructed in which the risk score, derived from the expression profiles of these genes, along with gender, emerged as robust independent predictors of patient outcomes in GC. Enrichment analysis linked high-risk patients to synaptic membrane pathways and low-risk to the CMG complex pathway. Tumor immune infiltration analysis revealed correlations between risk scores and immune cell abundance, suggesting a relationship between NAD+ metabolism and immune response in GC. The prognostic significance of our identified genes was validated by qRT-PCR, which confirmed their upregulated expression in GC tissue samples.

Conclusion

In this study, seven NAD+ metabolism-related markers were established, which is of great significance for the development of prognostic molecular biomarkers and clinical prognosis prediction for gastric cancer patients.

Introduction

Gastric cancer (GC) remains a prevalent and deadly disease worldwide, consistently ranking high in terms of incidence and mortality among various malignancies. In 2020 alone, there were over one million new cases and approximately 769,000 deaths attributable to GC (Sung et al., 2021). The disease is notorious for its deceptive onset, with early-stage patients frequently exhibiting no obvious symptoms. This subtle presentation makes early detection challenging, and consequently, many individuals are diagnosed at advanced stages, often when the cancer has already metastasized to distant organs (Wagner et al., 2017). Current therapeutic options are hindered by their limitations, frequently resulting in suboptimal patient outcomes. The intrinsic heterogeneity of tumors compounds this issue, leading to variable treatment responses across the patient population. The harsh toxicity and side effects associated with chemotherapy regimens are intolerable for many, leaving a subset of patients—especially those with advanced or metastatic GC—ineligible for curative surgery or unable to achieve clean surgical margins (R0 resection). In light of these profound challenges, there is a critical need to identify and establish more definitive prognostic biomarkers and predictive models for GC. Such advancements hold great promise in refining the prognostic assessment of GC patients. They serve as vital tools to inform clinical decisions, optimize treatment strategies, and ultimately enhance survival outcomes. The continued pursuit of biomarker discovery and model development is integral to the evolution of personalized medicine in the fight against GC.

Nicotinamide adenine dinucleotide (NAD+) is an essential auxiliary regulatory factor in cells, which is involved in cellular energy metabolism, protein post-translational modification, DNA repair, cell stress adaptation, and other biological processes. NAD+ metabolic disorders are closely related to various physiological changes and pathological damage, and it is also closely linked to the aging process of the body, metabolic and neurodegenerative diseases, and the development and progression of various cancers (Okabe et al., 2019; Xie et al., 2020). NAD+ levels decline steadily with age, leading to metabolic changes and increased disease susceptibility. Restoring NAD+ levels in elderly or diseased animals can promote health and prolong life (Rajman, Chwalek & Sinclair, 2018). In the classic remedial pathway, Nicotinamide phosphoribosyl transferase (NAMPT) As a rate-limiting enzyme, NAM is converted into nicotinamide mononucleotide (NMN), which is then converted into NAD by nicotinamide mononucleotide adenylyl transferase (NMNAT) (Sharif et al., 2019). It has been reported that NAMPT is highly expressed in various tumors such as GC, breast cancer, pancreatic cancer, and prostate cancer to promote glycolysis, proliferation, survival, invasion, metastasis, and chemotherapy resistance of tumor cells (Zhang et al., 2019; Lee et al., 2018; Qiu et al., 2017; Sampath et al., 2015). The relationship between gastric cancer (GC) and NAD+ is significantly influenced by sirtuins (SIRTs), which are a group of NAD+-dependent histone deacetylases. These enzymes play a crucial role in regulating metabolic pathways within the body (Poniewierska-Baran, Warias & Zgutka, 2022). While substantial research has been conducted on the prognosis of GC, the study of NAD+ metabolism-related genes in GC is still in its infancy. Therefore, it is imperative to explore the correlation between NAD+ metabolism-related genes and the prognosis of GC patients. This could provide new prognostic biomarkers and therapeutic targets, offering novel avenues for the treatment of GC.

In this study, intersection genes were identified by comparing the differentially expressed genes (DEGs1) between GC and paraneoplastic samples and the DEGs2 between NAD+ high and low score groups. Based on this, a risk model was constructed by Cox analysis and Least absolute shrinkage and selection operator (LASSO) analysis. After that, independent prognostic factors were obtained by Cox analysis, and a nomogram was developed to predict the survival rate of GC patients. Moreover, immune infiltration analysis was performed to explore the relationship between the prognostic genes and differential immune cells. These findings can provide a theoretical basis for NAD+ as prognosis prediction and treatment strategies for patients with GC.

Materials and Methods

This article has been posted as a preprint on Research Square: Yu Xing, Zili Zhang, Wenqing Gao et al. Construction of a prognostic model for gastric cancer patients based on NAD+ metabolism-related genes and tumor microenvironment analysis, 05 April 2023, PREPRINT (Version 1) available at Research Square (https://doi.org/10.21203/rs.3.rs-2711008/v1).

Data source

GC related RNA-seq dataset (TCGA-GC) were downloaded from The Cancer Genome Atlas database (TCGA, https://portal.gdc.cancer.gov), and clinical data were downloaded from the cbioportal database (https://www.cbioportal.org/). The TCGA-GC included 375 GC tissues and 32 paraneoplastic tissues, of which 354 GC samples with overall survival (OS) information. In addition, the GSE84437 dataset was downloaded from the Gene Expression Omnibus database (GEO, http://www.ncbi.nlm.nih.gov/geo/), which included sequencing data of 433 GC samples. Moreover, the 51 NAD+ metabolism-related genes (NMRGs) were sourced from the pathway of hsa00760 in the Kyoto Encyclopedia of Genes and Genomes database (KEGG, https://www.kegg.jp/ or https://www.genome.jp/kegg/) and the pathway of R-HSA-196807 in the Reactome database (https://reactome.org) (Li et al., 2022).

Differential expression and enrichment analysis

This study excavated the differentially expressed genes (DEGs1) between GC tissues and paraneoplastic tissues through the “limma” R package (version 3.52.1) with adjust. P < 0.05 and |logFC| > 1 (Ritchie et al., 2015). Then heat map and volcano map of DEGs were plotted using the R package “pheatmap” (version 1.0.12) (Hu, 2021) and R package “ggplot” (version 3.3.6) (Coleman, Bose & Mitra, 2023). Moreover, NMRGs were crossed with DEGs1 to obtain differentially expressed NMRGs (DE-NMRGs). At last, the Gene Ontology (GO) and KEGG enrichment analysis for DE-NMRGs was performed via R package “clusterProfiler” (version 4.0.2) (Wu et al., 2021). The threshold value was P < 0.05.

Correlation analysis of high and low score groups

Gene set variation analysis (GSVA) was performed on GC samples with OS in the TCGA-GC using NMRGs as the background gene set, and GC patients were divided into high and low score groups according to the median value of the scores. Then, the DEGs2 were obtained in the high and low score groups by differential analysis, which were intersected with DEGs1 to gain the intersection genes. In addition, the survival analysis of GC patients in the high and low score groups was performed using progression-free survival (PFS), and the genes were ranked according to the logFC. Afterthat, the gene set enrichment analysis (GSEA) analysis was performed for the high and low score groups. At last, the proportion of 22 immune cells in the high and low score groups was investigated by CIBERSORT algorithm, which was also compared between high and low score groups by the Wilcox test.

Construction and validation of a risk model

The GC patients with OS in the TCGA-GC were used as the training set. The univariate Cox (“survival” package, version 3.2–13) and LASSO regression (“glmnet” package, version 4.1–3) (Engebretsen & Bohlin, 2019) analysis based on intersection genes were deployed to select prognostic genes for construction of risk model in the training set. The risk score was calculated according to the formula. Riskscore = h0(t) * exp (β1X1 + β2X2 + … + βnXn), where the β represented the genes regression coefficient and the h0(t) represented the baseline hazard function. Additionally, GC patients were separated into two risk subgroups (high risk and low risk) based on the median of risk score. Moreover, the Kaplan-Meier (K-M) curves and risk curves was used to evaluate survival difference in different risk group, and the receiver operating characteristic (ROC) curve was generated to demonstrate the predictive efficiency of risk model. In addition, the risk model was further validated in the GSE84437 dataset by plotting risk curves, K-M curves, and ROC curves.

Independent prognostic analysis and construction of a nomogram

The risk score and clinical pathological factors were enrolled into univariate Cox and multivariate Cox analysis to obtain independent prognostic factors. Afterthat, the nomogram comprising the independent prognostic factors was drawn using R package “rms” (version 6.1–0) (Ramadan et al., 2020) to predict survival at 1, 3, and 5 years in GC patients. Moreover, the calibration curves were plotted by “regplot” package (version 1.1) to appraise the precision and reliability of the nomogram predictions (Zhang et al., 2018).

Functional enrichment analysis and immune infiltration analysis in different risk groups

To further investigate the signaling pathways and potential biological mechanisms in the high and low risk groups, GO and KEGG enrichment analyses were performed by GSEA. The threshold was set at P < 0.05 and |NES| > 1. In addition, the proportion of 28 immune cells in the high and low risk groups were analyzed by single sample gene set enrichment analysis (ssGSEA) algorithm using the R package “GSVA” (Hanzelmann, Castelo & Guinney, 2013), and the immune cells with significant differences were obtained by rank sum test. Finally, the relationships between immune cells and risk score were calculated by R package “ggplot”.

The mRNA expression of prognostic related genes was verified by experiment

In this study, tissue samples were collected from three patients with GC. All GC patients were from the Tianjin Third Central Hospital and signed informed consent. This study was approved by the Ethics Committee of Tianjin Third Central Hospital (SZX-20230222). The control group was paraneoplastic tissue samples, and the tumor group was GC tissue samples. The real-time reverse transcription-PCR (qRT-PCR) was carried out according to the basic operation steps (Tianjin Key Laboratory of Extracorporeal Life Support for Critical Diseases), and all consumables and solutions (Takara Company) were sterilized to prevent degradation and contamination of RNA during extraction (260/280 the value was between 1.8–2.0). Samples are placed in liquid nitrogen immediately after collection and subsequently transferred to −80 °C for storage (within 2 h) for less than 2 months. The integrity of RNA is determined by nucleic acid gel electrophoresis. The concentration and purity of total RNA in each sample were determined using a NanoDrop2000 spectrophotometer. The PCR machine uses the ABI 7500. At low temperature, the tissue was ground into homogenate, and the total RNA (2 µg) of the tissue was extracted by the Trizol method. The cDNA was obtained by reverse transcription reaction according to the operating guide of the reverse transcription kit. Related gene primers were diluted to 10 uM with DEPC water, then the DNA amplification reaction system was constructed according to the instructions of the gene amplification kit. Afert that, the qRT-PCR was conducted in and the reaction procedure was set as follow: pre-denaturation at 94 °C for 30 s, denaturation at 94 °C for 10 s, annealing at 65 °C for 15 s, extension at 72 °C for 10 s, and amplification for 40 cycles. Finally, the gene expression level was calculated by the 2−ΔΔCT method. Primer sequences for the genes can be seen in Table S1 (Guangzhou Ruibo Biotechnology Co., Ltd., Jiangsu, China).

Results

Enrichment analysis of 10 DE NMRGs

Upon analyzing the data, we identified 5,018 DEGs1 between GC and paraneoplastic tissues were obtained, with 3,964 up-regulated genes and 1,054 down-regulated genes (Figs. 1A and 1B, Table S2). By overlapping these DEGs with NMRGs), we isolated 10 DE-NMRGs (Fig. 1C). These DE-NMRGs were enriched in 152 GO Biological Process (BPs), 38 GO molecular function (MFs), and 13 KEGG pathways. For example, DE-NMRGs might influence the development and progression of GC through various pathways, including NAD biosynthetic process, nicotinamide nucleotide biosynthetic process, positive regulation of immune response to tumor cell, positive regulation of immune effector process, JAK-STAT signaling pathway, nicotinate and nicotinamide metabolism and nucleotide metabolism, etc. (Figs. 1D and 1E, Tables S3 and S4).

Figure 1 Enrichment analysis of DE NMRGs.

(A) Volcano plot and (B) heat map of DEGs between GC and paraneoplastic tissues in the TCGA dataset. (C) Thevenn diagram of NMRGs and DEGs between GC and paraneoplastic tissues. (D) GO enrichment analysis of DE NMRGs. (E) KEGG enrichment analysis of DE NMRGs.

Correlation analysis of GSVA

The 354 GC patients were divided into the high score group (n = 177) and low score group (n = 177). Analysis revealed 2,956 DEGs2 between high and low score groups (Table S5), and a total of 1,001 intersection genes were identified at the intersection (Fig. 2A). Survival analysis presented in Fig. 2B indicated that patients within the high score group exhibited a decreased survival rate compared to their low score counterparts. Moreover, GSEA highlighted enrichment in 1,501 GO annotations and 80 KEGG pathways (Tables S6 and S7). The top five GO and KEGG items enriched by high score (NES > 0) and low score (NES < 0) patients were visualized in Figs. 2C and 2D, including arrhythmogenic right ventricular cardiomyopathy, cell adhesion molecules, circadian entrainment, cell–cell adhesion via plasma-membraneadhesion molecules, intrinsic component of synaptic membrane etc. Importantly, the CIBERSORT algorithm analysis revealed differential proportions of immune cells including native B cells, memory B cells, regulatory T cells (Tregs), marcophage M0, resting mast cells, activated mast cells, and neutrophils. Of note, memory B cell levels were nearly undetectable in the majority of samples.

Figure 2 Differential expression and enrichment analysis.

(A) Heat map of GC-NM DEGs between high and low scoring groups. (B) Comparison of survival rates between high and low scoring groups. (C) GO and (D) KEGG enrichment analysis of high-score and low-score patients.

Construction and validation of a risk model

To mine the NMRGs relevant to the overall survival (OS) of GC patients, we incorporated the 1,001 intersection genes obtained above into a univariate Cox analysis in the training set, resulting in 387 genes associated with prognosis (Fig. 3A, Table S8). The above 387 genes were further integrated into the LASSO analysis, which identified 16 characteristic genes, namely NPY1R, NPR3, CCNA1, AFF2, DNAJB13, SNCG, OLFM3, CFHR3, CST2, CLDN10, DAO, THPO, CIDEA, NCCRP1, ONECUT1, UPK1B (Fig. 3B). Subsequently, a multivariate Cox analysis on the 16 characteristic genes identified seven prognostic genes: DNAJB13, CST2, THPO, CIDEA, ONECUT1, UPK1B, SNCG (Fig. 3C). Next, we computed the risk score for each GC patient in the training set and classified them into two risk subgroups (high risk and low risk) relying on the median value. Figure 3D revealed the distribution of ranked risk score and survival status for each patient in the training set. The expression heat map showed that DNAJB13, CST2, THPO, CIDEA, ONECUT1, UPK1B, and SNCG were highly expressed in patients with higher risk scores (Fig. 3E). The K-M curve demonstrated that patients in high risk group had noteworthy poorer survival than those with low risk group (Fig. 3F). We graphed ROC curves to check the predictive efficiency of the risk model. The area under the curve (AUC) values risk model in the training set were 0.699 (1 year), 0.728 (3 year), and 0.712 (5 year), indicating a decent accuracy (Fig. 3G). In addition, a comparable trend was verified in the external validation set (Figs. 4A–4D). Above results revealed that the risk signature was a valid survival predictor for GC patients.

Figure 3 Construction of a prognostic model in the training set.

(A) The forest map of univariate COX survival analysis. (B) Feature genes were screened by lasso regression analysis. (C) The forest map of multivariate COX survival analysis. (D) The risk curve for high-risk and low-risk groups. (E) Risk score heat map of prognostic biomarkers. (F) K-M survival curve for high and low risk groups. (G) ROC curves for 1-, 3- and 5-year survival in the training set.

Figure 4 Validation of the prognostic model in the validation set.

(A) Validate the vital status of patients in the high-risk and low-risk groups of the GSE84437 set. (B) Heat maps of key gene expression profiles in the validation set. (C) The K-M survival curves of high and low groups in the the validation set. (D) ROC curves for 1, 3, and 5-year survival of the risk groups of the validation set.

The risk score and gender were independent prognostic factors

Additionally, subsequent Cox regression analysis identified risk score and gender as independent predictors of prognosis for GC patients (P < 0.05) (Figs. 5A and 5B). A nomogram incorporating these independent prognostic predictors was constructed (Fig. 5C). The calibration curves proved that the capability of the nomogram in predicting survival at 1, 3, and 5 years in GC patients was satisfactory (Fig. 5D).

Figure 5 The risk score and gender were independent prognostic predictors.

(A) Forest plots of univariate COX and (B) multivariate COX. (C) The nomogram for predicting 1-, 3-, and 5-year survival of GC patients in the training set. (D) Calibration curve of the nomogram.

Functional enrichment analysis of high and low risk groups

In the functional enrichment analysis, a total of 872 GO items and 53 KEGG pathways were enriched (Tables S9 and S10). The top five GO and KEGG items enriched by high risk (NES > 0) and low risk (NES < 0) were shown in Figs. 6A and 6B. In the high-risk group, processes such as homophilic cell adhesion via plasma membrane adhesion molecules, multicellular organismal signaling, and postsynaptic membrane activity were activated (NES > 0). Conversely, in the low-risk group, pathways such as the CMG complex, DNA strand elongation involved in DNA replication, PPAR signaling pathway, and complement and proteasome assembly were activated.

Figure 6 Functional enrichment analysis of high and low risk groups.

(A) GO enrichment analysis for high and low risk groups. (B) KEGG enrichment analysis for high and low risk groups.

Relationships between risk score and immune cell

There were 11 types of immune cells that were significantly different between risk groups. Among them, activated CD4 T cells, activated CD8 T cells, type 17 T helper cells were highly expressed in high risk group, and central memory CD4 T cells, T follicular helper cells, activated B cells, immature B cells, natural killer cells, plasmacytoid dendritic cells, immature dendritic cells, and mast cell were low expressed in high risk group (Fig. 7A). In addition, seven immune cells (activated CD4 T cell, activited CD8 T cell, activated dendritic cell, CD56 bright natural killer cell, gamma delta T cell, memory B cell, plasmacytoid dendritic cell) were significantly correlated with the risk score (Fig. 7B). This correlation suggests that the risk score might either directly or indirectly influence the development of GC through its impact on immune cell dynamics.

Figure 7 Relationships between risk score and immune cell.

(A) Boxplot of immune cell distribution between high and low risk groups. (B) Graph of correlation between immune cells and risk score. *P < 0.05; **P < 0.01; ***P < 0.001.

Expression of prognostic genes

In order to further verify the expression of the prognostic gene, we studied the relative expression by qRT-PCR. The results showed that the expressions of DNAJB13, CST2, THPO, CIDEA, ONECUT1, UPK1B, and SNCG were significantly up regulated in GC tissues compared with those of paraneoplastic tissues (Fig. 8).

Figure 8 Expression of prognostic genes.

Comparison of DNAJB13, CST2, THPO, CIDEA, ONECUT1, UPK1B, and SNCG mRNA expression between paraneoplastic tissues (control group) and GC tissues (tumor group), *P < 0.05; **P < 0.01; ***P < 0.001; ****P < 0.0001.

Discussion

Studies have shown that cancer cells, known for high glycolysis levels, produce large amounts of NAD+ (Yong, Cai & Zeng, 2023). NAD+ is crucial in energy metabolism and important cellular signaling processes like mono-ADP and poly-ADP ribosylation, as well as NAD+-dependent protein deacetylation (Diefenbach & Burkle, 2005). These signaling events are intimately linked to GC progression. Thus, our study, based on TCGA and GEO data, constructed an NMRG-related risk model for GC and explored its correlation with the tumor microenvironment, providing a theoretical basis for GC treatment. Our research pioneers a prognostic model grounded in NMRGs for GC, a domain yet to be thoroughly explored. This innovative approach merges molecular biology with clinical prognostics, setting a novel paradigm for understanding and managing GC. The core novelty of our study lies in utilizing specific NMRGs as prognostic biomarkers for GC. This method could open up new possibilities for improving clinical outcomes and underscores our dedication to advancing GC research with significant and impactful insights.

The univariate Cox, LASSO, and multivariate Cox analyses in the study demonstrated strong model robustness and interpretability, effectively controlling overfitting and facilitating variable selection. These methods can be applied to clinical decision-making, enabling more accurate and effective identification and validation of biomarkers associated with patient prognosis. Based on these analyses, seven prognostic gene (DNAJB13, CST2, THPO, CIDEA, ONECUT1, UPK1B, SNCG) were screened out for constructing risk model. CST2 expression is significantly up-regulated in GC samples, which enhances the growth, migration, and invasion of GC cells by regulating epithelial-mesenchymal transition (EMT) and TGF-β1 signaling pathways, leading to poor prognosis in GC patients (Zhang et al., 2020). Thrombopoietin (THPO), involved in megakaryocyte proliferation and maturation, is overexpressed in GC. This overexpression contributes to poor prognosis by impacting cell viability, invasion, and migration (Zhou, Su & Dai, 2020). In recurrent GC, patients elevated expression of UPK1B is associated with poorer prognosis (Zhang et al., 2020). This adverse outcome may be due to UPK1B significantly influencing the expression of key genes in the Wnt/β-catenin signaling pathway in cancer cells (Wang et al., 2018). Moreover, SNCG expression in gastric adenocarcinoma is considerably higher than in non-tumorous adjacent gastric tissues, correlating with the depth of invasion and lymph node metastasis (Zheng et al., 2014). These findings indicate that higher expression levels of these genes in GC are linked to worse patient prognosis, consistent with the results of our study and enhancing the credibility of our conclusions. In addition, among these seven genes, we first found that DNAJB13, CIDEA, and ONECUT1 were associated with the prognosis of GC patients. Although these genes have not been reported in GC, we were surprised to find that these three genes have been reported in colon cancer. Studies have shown that compared with normal colonic mucosa, the expression of brown fat-associated protein (CIDEA) in colorectal tumors increased significantly. The brown fat-like phenotype characterized by CIDEA expression may be due to the infinite plasticity of cancer stem cells, which can differentiate into multiple cell lines with various phenotypes according to gene mutations and microenvironment factors. These cells undergo carcinogenesis under the influence of multiple factors (Alnabulsi et al., 2019). ONECUT1 is highly expressed in colorectal cancer and has a poor prognosis. The mechanism may be that ONECUT1 promotes the proliferation and tumor growth of colorectal cancer cells and promotes liver metastasis by endogenic resistance to anoikis (Jiang et al., 2019). The high expression of DNAJB13 in colon cancer tissues also confirmed that DNAJB13 is an independent prognostic risk factor for colon cancer patients (Lu et al., 2020). The results of the above three genes are consistent with our experiments. Therefore, we preliminarily speculate that these genes may be independent risk factors in GC.

This study is the first to construct a prognostic nomogram model related to NMRGs in GC. Compared with the traditional TNM staging system, the nomogram model based on multivariate analysis has been proven to obtain more accurate prognostic prediction in various cancer patients (Ruan et al., 2021; Tonello et al., 2021). A study utilized data from the SEER database to develop a nomogram for predicting cancer-specific survival in patients with metastatic clear cell renal carcinoma (mccRCC) (Huang et al., 2022). This model included variables like age, gender, race, T-stage, N-stage, types of metastases, and treatments like chemotherapy, radiotherapy, and surgery. The model was validated and found to be more accurate than the traditional TNM staging system in predicting the survival of mccRCC patients. In a study involving patients with locally advanced breast cancer (LABC), two prediction models were developed to predict OS and cancer-specific survival (CSS) (Yin et al., 2022). These models integrated several variables and demonstrated a higher accuracy in predicting survival outcomes for LABC patients compared to the TNM staging system. The models were also used to guide treatment plans for these patients.Studies have shown that the NMRGs risk score prediction model is closely related to the prognosis of liver cancer, and the high risk group was significantly worse than the low-risk group (Wang et al., 2022). This is consistent with our experimental results. The simple and easy-to-use nomogram prediction model constructed in this study can help clinicians make more accurate survival predictions, identify high-risk patients, and conduct early individual intervention and treatment. In terms of GC prognosis models, many studies may use larger and more diverse data sets, which can enhance the generalization ability and reliability of the model. Feature selection techniques based on machine learning, such as LASSO regression or tree-based algorithms, are also used, which can help us extract more representative features and improve the prediction performance of the model (Yang et al., 2023; Mak et al., 2022; Yang et al., 2022; Liu et al., 2023). Previous scholars utilized the database to construct an iron metabolism-related prognostic model for GC, the risk model developed on a TCGA dataset yielded AUCs of 0.627, 0.643, and 0.631 for 1, 3, and 5 years, respectively (Liu et al., 2023). In comparison, the risk model we delineated within the TCGA dataset demonstrated a more robust AUC profile, with respective AUCs of 0.699, 0.728, and 0.712 for 1, 3, and 5 years. This indicates a significant enhancement in both the sensitivity and specificity of our model. These enhancements not only attest to the superior predictive capability of our model but also underscore its robust applicability and consistency across varying clinical contexts. Furthermore, the identification of independent prognostic factors, namely risk score and gender, suggests that their prognostic impact remains steadfast irrespective of other variables. This underscores the potential clinical utility of our risk score and reaffirms the substantial value of our research. Our findings indicate that this risk score could reliably contribute to the prognostic assessment in clinical practices, offering a broadened scope for its applicability in individualized patient care. Future investigations have the potential to refine the clinical prognostic assessment for GC patients by integrating these two key factors, thereby enhancing the precision of prognostic evaluations. The inclusion of these independent prognostic markers within our nomogram has notably augmented its accuracy and clinical pertinence in forecasting outcomes for GC patients. This improvement sheds light on the instrument’s inherent value in personalized patient care and informed decision-making. Our intricately developed nomogram serves as an invaluable predictive tool with promising applications in clinical settings, aiding in the nuanced management of GC patient care. Anticipated to influence the trajectory of forthcoming research, these findings empower clinicians to tailor treatment regimens to the distinct pathophysiological profiles of individual patients’ diseases. The advent of our model is poised to support healthcare professionals in delivering precision medicine, elevating the standard of care for those battling gastric cancer.

We found that there were 11 types of immune cells that differed significantly between the risk groups. Moreover, tumor-associated immune cells have a positive effect on the prognosis of GC (Yang et al., 2021). Tumor-associated lymphocytes have a positive effect on the prognosis of GC. Dendritic cells (DCs) and natural killer (NK) cells initiate specific immune responses to tumor cells (Chen et al., 2014). The decrease of subsets of DC and NK and cytokines may be one of the reasons for the decrease of DC and NK function in tissues and peripheral blood of patients with GC. Human NK cells can be divided into two subsets according to the expression of CD56 and CD16. The H₂O₂ produced in the tumor microenvironment is inversely proportional to the infiltration of CD56, which may be due to its preferential induction of cell death. It further explains one of the mechanisms behind NK cell dysfunction often observed in the tumor microenvironment (Izawa et al., 2011). In addition, the T cells are an important part of tumor infiltrating lymphocytes (TILs), and T cell-mediated adaptive immunity is considered to play an important role in anti-tumor immunity. T cell subsets are composed of CD8, CD4 and other cells. These lymphocytes can infiltrate the matrix and tumor cells to regulate the host’s immune response to tumor cells (Oya, Hayakawa & Koike, 2020), Up-regulated PD-L1 or CTLA-4 expression inhibits these anti-tumor immune responses in some tumor microenvironments.

By GO enrichment analysis, we found that multicellular organismal signaling were significantly enriched in GO. Multicellular organismal signaling is a complex molecular network that transmits information between different cells in multicellular organisms to coordinate a variety of physiological processes, including cell growth, differentiation, metabolism, cell death, and immune responses (Hanahan & Weinberg, 2011; Saltiel & Kahn, 2001; Tsokos, 2011). The research on “multicellular organismal signaling” in relation to GC indicates a significant connection. Studies have found that NEK7, a protein involved in cell proliferation, is highly expressed in GC cells and is associated with poor prognosis (Li et al., 2021). NEK7 expression is related to advanced GC staging and is implicated in immune cell infiltration in GC. Furthermore, NEK7 has been identified as a key factor in several signaling pathways, including cell-cell adhesion via plasma membrane adhesion molecule pathways and multicellular organismal signaling pathways, which are crucial in regulating multicellular signaling and intracellular proliferation-related pathways in GC. This suggests that components of multicellular organismal signaling, like NEK7, play an important role in the progression and severity of GC.

The cGMP-PKG signaling pathway plays a significant role in the regulation of GC. This study highlights the role of Type II cGMP-dependent protein kinase (PKG II) in inhibiting EGF-triggered signal transduction of the MAPK/ERK-mediated pathway in GC cells (Wu et al., 2012). This suggests that the cGMP-PKG pathway, through its components like PKG II, might have a tumor-suppressing effect on GC cells by inhibiting certain pathways that are crucial for cancer cell migration and proliferation. These findings suggest that the cGMP-PKG signaling pathway could be a potential target for therapeutic interventions in GC, offering new avenues for treatment and understanding of the diseases.

However, we must acknowledge certain limitations in our research. Firstly, the data utilized were sourced from public databases, which may introduce heterogeneity. Secondly, the mechanisms of action of these prognostic genes in GC require further experimental validation, and the clinical application prospects of our prognostic model remain to be confirmed. Thirdly, the sample size of the datasets used in our study may affect the accuracy of our results and potential biases, and the application prospects in large-scale clinical settings also necessitate verification with a more comprehensive and larger number of samples. Lastly, our study is rooted in bioinformatics; therefore, extensive in vitro and clinical investigations are crucial for substantiating our findings. Our future work will focus on more extensive experimental and clinical studies to continue exploring the significance of these prognostic genes in GC.

Conclusion

In this study, we identified prognostic genes and independent prognostic factors related to NMRGs in GC for the first time through bioinformatics methods, providing a new idea for the clinical treatment of GC.

Supplemental Information

Supplemental Information 1 The primer sequence of the gene.

Supplemental Information 2 Genes for GC and paraneoplastic tissues.

A total of 5,018 DEGs1 between GC and paraneoplastic tissues were obtained

Supplemental Information 3 Biological Process (BPs), Molecular Function (MFs).

they were enriched to 152 GO Biological Process (BPs), 38 GO Molecular Function (MFs)

Supplemental Information 4 KEGG pathways.

13 KEGG pathways

Supplemental Information 5 DEGs2 between high and low groups.

There were 2,956 DEGs2 between high and low score groups

Supplemental Information 6 1,501 GO items.

Supplemental Information 7 80 KEGG pathways.

Supplemental Information 8 387 prognostic genes.

Supplemental Information 9 GO items.

A total of 872 GO items were enriched.

Supplemental Information 10 KEGG pathways.

A total of 53 KEGG pathways were enriched.

Supplemental Information 11 PCR.

Supplemental Information 12 PCR procedure.

Supplemental Information 13 MIQE_checklist.

We thank the TCGA and KEGG databases for providing meaningful data sets.

Additional Information and Declarations

Competing Interests

Author Contributions

Human Ethics

Data Availability

The authors declare that they have no competing interests.

Yu Xing performed the experiments, analyzed the data, prepared figures and/or tables, authored or reviewed drafts of the article, and approved the final draft.

Zili Zhang performed the experiments, prepared figures and/or tables, and approved the final draft.

Wenqing Gao conceived and designed the experiments, analyzed the data, prepared figures and/or tables, and approved the final draft.

Weiliang Song conceived and designed the experiments, analyzed the data, authored or reviewed drafts of the article, and approved the final draft.

Tong Li conceived and designed the experiments, analyzed the data, authored or reviewed drafts of the article, and approved the final draft.

The following information was supplied relating to ethical approvals (i.e., approving body and any reference numbers):

The Ethics Committee of Tianjin No. 3 Central Hospital granted ethical approval to conduct research in its facilities.

The following information was supplied regarding data availability:

The original measurements are available in the Supplemental File. Additional tables are available at GitHub: https://GitHub.com/kunlunyou/doc/tree/main.

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
