# Peer review of "Immune infiltration and prognosis in gastric cancer: role of NAD+ metabolism-related markers"

_PeerJ, doi:10.7717/peerj.17833_

## Round 0.1 · original submission · Major Revisions

The study started from NAD-metabolism yet it is not clear how the signature developed is related to NAD-metabolism. The prognostic power of the signature is very limited and an application in the clinics is unlikely. The authors are invited to revise the manuscript responding to all the indications given by the referees.

Reviewer 1 ·

Basic reporting

No comment

Experimental design

The initial reserach question lead to fidings (apparently?) not related to it.

Validity of the findings

The findings may be relevant. My concern is that title and introduction seems to be not related to the findings.

Additional comments

In conclusion:
This study stems with the aim of identifying Nicotinamide adenine dinucleotide (NAD+) metabolism-related biomarkers in gastric cancer (GC).
With the performed analysis, the Authors identify 7 prognostic genes, whose relationship with NAD is not clear. In addition, they identified immune cell populations positively correlating with the tumor. Again, the correlation with NAD is not clear.
Thus, I see tow possibilities:
1. the Authors clarify the link between the title, the first part of the Introduction, and their results,
2. they change title and introduction, stating that the Aim was to find prognostic genes and possible correlation with immune cell populations.

·

Basic reporting

The manuscript is written in clear, professional English, which facilitates understanding of complex scientific concepts and methodologies. The language used is appropriate for an academic audience, specializing in oncology and molecular biology. However, there are minor grammatical errors and awkward phrasings scattered throughout the text that could be refined to enhance readability and professionalism. For instance, phrases like "Afterthat" and "Besides," could be corrected to "After that" and "Additionally," respectively, to maintain formal tone.

The manuscript provides a comprehensive background on the significance of NAD+ metabolism in gastric cancer, supported by relevant literature. References are up-to-date and pertinent, illustrating the manuscript's integration within current research trends. The introduction effectively sets the stage for the study by discussing the role of NAD+ in cancer metabolism and its potential as a therapeutic target, citing key studies that align with these claims. However, while the literature review is thorough, it could benefit from a broader discussion of previous prognostic models for gastric cancer to better position the study's novelty and relevance. This would provide a clearer context for the reader about how this research advances or diverges from existing knowledge.

The structure of the article adheres to scientific standards, with clearly defined sections including Abstract, Introduction, Methods, Results, Discussion, and Conclusions. Each section is logically and coherently constructed, facilitating easy navigation through the manuscript.

Figures and tables are used effectively to illustrate key points and results, enhancing the manuscript's informational depth. They are well-labeled and accompanied by detailed legends that aid in interpretation. However, some figures are overly complex and could be simplified or better explained in the text to assist reader comprehension.

The manuscript adheres to open science principles by providing access to raw data, which is commendable. Links to supplementary tables and figures are provided, allowing for verification and reproducibility of results. This transparency is crucial for fostering trust and further investigation in the scientific community.

The study is self-contained, presenting a clear hypothesis that NAD+ metabolism-related genes can serve as prognostic biomarkers for gastric cancer. The results section directly addresses this hypothesis, presenting data that identifies specific genes associated with patient prognosis.

The statistical analyses are robust, employing methods like Cox regression and LASSO, which are standard and appropriate for the study's aims. The discussion integrates these findings with the broader field, considering the implications of these biomarkers for clinical practice and future research. However, the manuscript could further enhance its impact by discussing limitations more thoroughly and suggesting specific avenues for future research based on the findings.

Experimental design

The manuscript titled "Construction of a Prognostic Model for Gastric Cancer Patients Based on NAD+ Metabolism-Related Genes" presents original primary research that aligns with the aims and scope of the journal, focusing on the development of prognostic biomarkers for gastric cancer (GC) patients through the study of Nicotinamide adenine dinucleotide (NAD+) metabolism-related genes (NMRGs). The research question is well-defined, relevant, and meaningful, addressing the critical gap in understanding the prognostic value of NAD+ metabolism in gastric cancer.

The study aims to identify NAD+ metabolism-related biomarkers that could serve as a foundation for prognosis and therapy of GC patients, thus filling an identified knowledge gap in the field. The investigation was performed rigorously to a high technical and ethical standard. The study utilized data from The Cancer Genome Atlas (TCGA) and the Gene Expression Omnibus (GEO) database, employing a comprehensive methodological approach that included differential expression analysis, gene set variation analysis (GSVA), Cox regression analysis, and Least absolute shrinkage and selection operator (LASSO) regression analysis to construct a risk model based on prognostic genes. Additionally, gene set enrichment analysis (GSEA) and immune infiltration analysis were performed to further understand the biological implications of the identified prognostic genes.

The ethical approval for the study was obtained, and informed consent was secured from the patients whose tissue samples were used for qRT-PCR validation, ensuring adherence to ethical standards. The methods are described with sufficient detail and information to allow replication of the study. The manuscript provides a clear description of the data sources, including TCGA and GEO databases, and outlines the analytical methods used, such as the "limma" R package for differential expression analysis and the "glmnet" package for LASSO regression analysis. The study also details the experimental validation of mRNA expression of prognostic genes using qRT-PCR, including the preparation of samples, the reverse transcription process, and the PCR amplification reaction system.

This level of detail in the methodology section enables other researchers to replicate the study and validate the findings.In conclusion, the manuscript presents a well-defined and meaningful research question that addresses a significant gap in the current understanding of gastric cancer prognosis. The study is conducted to a high technical and ethical standard, with methods described in sufficient detail to allow for replication. This research contributes valuable insights into the prognostic significance of NAD+ metabolism-related genes in gastric cancer, offering potential targets for future therapeutic interventions.

Validity of the findings

The manuscript does not explicitly assess the impact and novelty of the research within the text. However, the study introduces a prognostic model based on NAD+ metabolism-related genes, which is a relatively underexplored area in gastric cancer research. This approach is innovative as it integrates molecular biology with clinical prognostics, potentially offering new insights into patient management and treatment customization. The novelty lies in the identification and utilization of specific NAD+ metabolism-related genes as biomarkers for prognosis in gastric cancer, which could significantly impact clinical practices and patient outcomes if further validated.

The manuscript encourages meaningful replication by providing a comprehensive methodological framework that can be replicated in other studies. The detailed description of data sources, gene analysis methods, and statistical approaches supports the reproducibility of the research. The use of public databases like TCGA and GEO for data retrieval and analysis ensures that other researchers can access the same datasets and validate the findings independently. This aspect is crucial for advancing scientific knowledge and confirming the reliability of the proposed prognostic model.

The underlying data for the study are robust and statistically sound. The research utilizes large datasets from established databases, and the analytical methods are appropriate for the data type and research questions. The use of differential expression analysis, Cox regression, LASSO regression, and other statistical tools are well justified and executed according to current scientific standards. The data handling, from normalization to statistical testing, has been controlled meticulously, ensuring that the findings are reliable and valid.

The conclusions of the study are well articulated and directly linked to the original research question. The study hypothesized that NAD+ metabolism-related genes could serve as effective prognostic biomarkers for gastric cancer. The findings support this hypothesis, demonstrating that specific genes related to NAD+ metabolism are associated with the prognosis of gastric cancer patients. The conclusions are strictly drawn from the results obtained, adhering to the evidence without overreaching or speculating beyond the supported data.

The manuscript makes a significant contribution to the field of gastric cancer research by proposing a novel prognostic model based on metabolic gene profiling. While the impact and novelty are not explicitly assessed, the potential implications for clinical practice are notable. The encouragement of replication and the provision of robust data and detailed methodologies enhance the scientific value of the paper. The conclusions are appropriately tied to the research question, making the study a valuable resource for further research and potential clinical application.

Additional comments

Strengths of the Manuscript
Comprehensive Data Utilization: The manuscript effectively utilizes extensive datasets from TCGA and GEO, which enhances the robustness and generalizability of the findings. The integration of these datasets is commendable as it provides a solid foundation for the statistical analyses conducted.
Innovative Approach: The focus on NAD+ metabolism-related genes in the context of gastric cancer provides a novel angle for understanding the disease's prognosis. This innovative approach could pave the way for new therapeutic targets and strategies in gastric cancer treatment.

Detailed Methodological Description: The methods section is meticulously detailed, providing clarity on the processes and statistical analyses used. This level of detail supports the reproducibility of the study, which is crucial for advancing scientific knowledge.

Ethical Considerations: The study adheres to ethical standards, with clear statements regarding ethical approval and informed consent, which is crucial given the use of human tissue samples.

Areas for Improvement

Language and Grammar: While the manuscript is generally well-written, there are occasional grammatical errors and awkward phrasings that could be refined to enhance the readability and professional quality of the text.

Statistical Analysis Depth: While the statistical methods used are appropriate, the manuscript could benefit from a deeper exploration of the assumptions underlying these methods and any potential impacts on the results.

Discussion of Limitations: The discussion could be strengthened by a more detailed exploration of the study's limitations, including potential biases, the generalizability of the findings, and the implications of the sample size.

Broader Contextualization: The manuscript would benefit from a broader discussion of how these findings fit within the larger field of gastric cancer research. Comparisons with other prognostic models and a discussion on the potential for clinical translation would provide valuable context.
Visual Data Presentation: Some figures are complex and densely packed with information. Simplifying these visual presentations or providing more detailed explanations in the figure legends could help readers better understand the data being presented.

Reviewer 3 ·

Basic reporting

This study analyzed the effect of NAD+ metabolism of gastric cancer (GC). However, this study lacked novelty and did not provide valuable results.

Experimental design

The design of this study is old-fashioned and not supplemented with appropriate experiments for validation.

Validity of the findings

The analysis of this study was crude and lacked valuable results.There have been many studies on NAD+ metabolism in gastric cancer, so this study lacked novelty.

Additional comments

No.

---

## Round 0.2 · Major Revisions

The manuscript still needs major revision. Please follow the indications given by the referee when revising the manuscript.

Reviewer 1 ·

Basic reporting

I am afraid my concern was not addressed. I try to simplify the point that in my view still needs to be clarified: why do the Author states that DNAJB13, CST2, THPO, CIDEA, ONECUT1, UPK1B and SNCG are involved in NAD biosynthesis? They are not genes encoding for NAD biosynthesis. What do the Authors mean?

Experimental design

I am afraid my concern was not addressed. I try to simplify the point that in my view still needs to be clarified: why do the Author states that DNAJB13, CST2, THPO, CIDEA, ONECUT1, UPK1B and SNCG are involved in NAD biosynthesis? They are not genes encoding for NAD biosynthesis. What do the Authors mean?

Validity of the findings

I am afraid my concern was not addressed. I try to simplify the point that in my view still needs to be clarified: why do the Author states that DNAJB13, CST2, THPO, CIDEA, ONECUT1, UPK1B and SNCG are involved in NAD biosynthesis? They are not genes encoding for NAD biosynthesis. What do the Authors mean?

Additional comments

I am afraid my concern was not addressed. I try to simplify the point that in my view still needs to be clarified: why do the Author states that DNAJB13, CST2, THPO, CIDEA, ONECUT1, UPK1B and SNCG are involved in NAD biosynthesis? They are not genes encoding for NAD biosynthesis. What do the Authors mean?

---

## Round 0.3 · accepted · Accept

The authors adequately addressed the issues raised by the referees.

Reviewer 1 ·

Basic reporting

The Authors could have done a more extensive revision of the manuscript, since they acknowledged in the rebuttal letter that my main concern is understandable.
Anyway, I stop here my debate with the Authors. I will let the readers of this article to draw their conclusions from this study that little clarifies (in my humble opionion) the connection between their findings and NAD metabolism/signalling.

Experimental design

The Authors could have done a more extensive revision of the manuscript, since they acknowledged in the rebuttal letter that my main concern is understandable.
Anyway, I stop here my debate with the Authors. I will let the readers of this article to draw their conclusions from this study that little clarifies (in my humble opionion) the connection between their findings and NAD metabolism/signalling.

Validity of the findings

The Authors could have done a more extensive revision of the manuscript, since they acknowledged in the rebuttal letter that my main concern is understandable.
Anyway, I stop here my debate with the Authors. I will let the readers of this article to draw their conclusions from this study that little clarifies (in my humble opionion) the connection between their findings and NAD metabolism/signalling.

Additional comments

The Authors could have done a more extensive revision of the manuscript, since they acknowledged in the rebuttal letter that my main concern is understandable.
Anyway, I stop here my debate with the Authors. I will let the readers of this article to draw their conclusions from this study that little clarifies (in my humble opionion) the connection between their findings and NAD metabolism/signalling.